# Test–Retest Reliability in Automated Emotional Facial Expression Analysis: Exploring FaceReader 8.0 on Data from Typically Developing Children and Children with Autism

Zsófia Borsos [1,2,3,4,*], Zoltán Jakab [3], Krisztina Stefanik [4,5], Bianka Bogdán [3] and Miklos Gyori [3,4]

1 Doctoral School of Psychology, ELTE Eötvös Loránd University, 1053 Budapest, Hungary
2 Institute of Psychology, ELTE Eötvös Loránd University, 1053 Budapest, Hungary
3 Institute for the Psychology of Special Needs, ELTE Eötvös Loránd University, 1053 Budapest, Hungary; jakab.zoltan@barczi.elte.hu (Z.J.); bianka.bogdan@barczi.elte.hu (B.B.); gyori.miklos@barczi.elte.hu (M.G.)
4 HAS-ELTE 'Autism in Education' Research Group, 1053 Budapest, Hungary; krisztina.stefanik@barczi.elte.hu
5 Institute of Special Needs Education for People with Atypical Behavior and Cognition, ELTE Eötvös Loránd University, 1053 Budapest, Hungary
* Correspondence: borsos.zsofia@barczi.elte.hu

**Featured Application: The results of this study can potentially contribute to the further development of automated emotional facial expression analysis systems, to their usage in developing screening/diagnostic systems, and to the refinement of methodological practices of their usage in both applied and basic research in general.**

**Abstract:** Automated emotional facial expression analysis (AEFEA) is used widely in applied research, including the development of screening/diagnostic systems for atypical human neurodevelopmental conditions. The validity of AEFEA systems has been systematically studied, but their test–retest reliability has not been researched thus far. We explored the test–retest reliability of a specific AEFEA software, Noldus FaceReader 8.0 (FR8; by Noldus Information Technology). We collected intensity estimates for 8 repeated emotions through FR8 from facial video recordings of 60 children: 31 typically developing children and 29 children with autism spectrum disorder. Test–retest reliability was imperfect in 20% of cases, affecting a substantial proportion of data points; however, the test–retest differences were small. This shows that the test–retest reliability of FR8 is high but not perfect. A proportion of cases which initially failed to show perfect test–retest reliability reached it in a subsequent analysis by FR8. This suggests that repeated analyses by FR8 can, in some cases, lead to the "stabilization" of emotion intensity datasets. Under ANOVA, the test–retest differences did not influence the pattern of cross-emotion and cross-group effects and interactions. Our study does not question the validity of previous results gained by AEFEA technology, but it shows that further exploration of the test–retest reliability of AEFEA systems is desirable.

**Keywords:** autism spectrum disorder (ASD); automated emotional facial expression analysis (AEFEA); children; Noldus FaceReader 8.0; test–retest reliability

## 1. Introduction

Automated emotional facial expression analysis (AEFEA) technologies are aimed at processing digitized images of human faces in order to generate estimates of the intensities of emotional states displayed on the face. It is expected that these systems can complete this process faster and more cost-effectively than human agents [1–4], with higher reliability and with at least comparable validity [5]. Although there is an ongoing discussion about AEFEA systems' potentially intrinsic limitations—arising from estimating emotional content without taking into account the context in which the facial expression arises [6,7]—there is a rapidly growing interest in them. Beyond their use in academic research, there have been

numerous attempts to utilize AEFEA technologies in applied fields, such as researching consumer behavior, user experience, and marketing [5,8] and in diagnostic technologies in psychiatry [6,9,10]. While their validity has been widely investigated, no study has been published on their test–retest reliability; the latter is an equally important property of any tool for collecting data on human behavior. Thus, the present study empirically explored the test–retest reliability of a specific AEFEA system. As AEFEA technology seems to have great potential in screening and diagnostic technologies for neurodevelopmental conditions, we used data from typically developing children and children with autism spectrum disorder (ASD). Our study suggests that the test–retest reliability of Noldus FaceReader 8.0 (FR8; by Noldus Information Technology; Wageningen, The Netherlands) is very high but not perfect. Our findings do not question the validity of previous results gained by AEFEA technology, but they demonstrate that further exploration is needed to gain a better understanding of this key characteristic of AEFEA systems. Test–retest reliability showed highly similar patterns in the groups of children with ASD and with neurotypical development.

Ekman's Facial Action Coding System (FACS) [11] is a widely used framework for identifying emotional facial expressions (EFEs) and assessing their intensities by human coders based on the anatomical features of the face. It also serves as the foundation for making emotion estimations in several AEFEA technologies [12], as it is highly valid and reliable when used by human coders [13,14]. It can be elaborated precisely enough to be implemented in computational algorithms [7,15]. It defines action units (AUs) anatomically, that is, through different muscle movements of the face, to identify specific EFEs as constituted by their specific combinations. Intensity estimations for any emotional expression are derived from the intensities of their component AUs, that is, from the amplitudes of the relevant muscle movements.

At present, a few commercially available versions of AEFEA technology exist, based on the FACS, including the AFFDEX (developed by Affectiva, Boston, MA, USA, distributed by Affectiva and iMotions, Copenhagen, Denmark) [16–18]; the FACET (developed by Emotient, San Diego, CA, USA, distributed by iMotions, Copenhagen, Denmark) [19]; and the Noldus FaceReader (developed by VicarVision, Amsterdam, The Netherlands, distributed by Noldus Information Technology) [20]. See Dupré, Krumhuber, Küster, and McKeown's review [21] for others. Although all of these are based on FACS, they use various computational algorithms for face and feature (AU) detection [21]. There is intense simultaneous development in this area, which stretches beyond classical emotion theories and FACS. Analysis of facial expressions has been supplemented by the processing of a broader set of social signals [22]. Emotion identification has been extended beyond basic or primary emotions (fear, disgust, anger, surprise, happiness, sadness, and contempt, according to Ekman's 1978 framework) to include secondary emotions (e.g., boredom, interest, and confusion) [23]. Facial expressions have also been decomposed into sequences of distinct phases, and their symmetry and micro-expressions, and the dynamic patterns of differences between morphologically similar facial expressions, have been analyzed [4]. Input data from laboratory settings are increasingly replaced with data from "in-the-wild" scenarios [2,4,22,24]. In this study, we focus on FACS-based AEFEA systems, and, more specifically, on FaceReader 8.0.

Although the number of AEFEA algorithms has increased in recent years, only a few studies on their validity and reliability have been published in peer-reviewed journals, e.g., [3,5,21,25]. More studies are available in conference presentations and proceedings [15,26–29].

Validation of AEFEA systems is conducted on controlled, publicly available reference datasets of images of human faces with EFEs and corresponding emotion intensities, as judged by trained human coders. Of the algorithms based on Ekman's FACS, FaceReader, AFFDEX, and FACET have been best documented to reach high validity [3,5,20,25,30].

In regard to the FaceReader system specifically, three different versions of FaceReader have been validated on four different validated datasets of human expressions of basic emotions in total, three of which are publicly available. FaceReader 1.0 exhibited an

89% match with the reference emotion labels from the Karolinska Directed Emotional Faces (KDEF) dataset [20,30,31]. FaceReader 6.0 reached an 88% overall hit rate [5] when validated on the Warsaw Set of Emotional Facial Expression Pictures (WSEFEP) [32] and the Amsterdam Dynamic Facial Expression Set (ADFES) [33]. Human emotion recognition performance for these two datasets was 85% in the same study. For some action units, FaceReader did not reach the 0.7 level of agreement required for the FACS certificate for human coders; however, for other AUs, FaceReader showed high accuracy. It remarkably surpassed human coders in identifying neutral faces [34]. FaceReader 7 was validated [35] on the Standardized and Motivated Facial Expression of Emotion (SMoFEE) dataset [36], and its performance was compared with that of FaceReader 6.0 and of human coders found in earlier studies. Overall, FaceReader 7 performed well, as it correctly classified 79% and 80% of EFEs (in noncalibrated and calibrated conditions, respectively); however, it showed somewhat uneven performance across emotions and underperformed FaceReader 6.0 in some respects.

FaceReader is under continuous development; its most recent release is version 9.

There have been considerable efforts to study both the inter-rater reliability and the test–retest reliability of human facial emotion analysis in the FACS framework. Inter-observer agreement was studied in various ways, such as using spontaneous vs. posed facial images and focusing on all AUs vs. specific AUs [1,37]. The most-cited study on human–human inter-rater reliability within the FACS framework using spontaneous facial expressions in three independent laboratory studies is that of Sayette et al. [14], which reported good–excellent reliability measures.

While both the inter-rater reliability and the test–retest reliability of human emotion analysis within the FACS framework have been explored, in the case of FACS-based AE-FEA systems, perfect test–retest reliability was tacitly taken for granted (see below) and, instead, inter-system reliability studies were published. In most of these, the systems' performances were evaluated individually and sequentially against reference datasets, and then compared. In other studies, agreement between the different systems was directly assessed. Study outcomes are not easy to compare, because diverse kits of stimulus datasets and statistics were used. This is true of human–computer inter-system comparisons [3,5,10,34,35,38,39]. A smaller number of inter-system comparison studies have employed more sophisticated methods, where aspects of emotion dynamics, such as onsets, peaks, and offsets of emotions, or changes in action unit intensities, were also compared [1,20]. Overall, the results indicate that AEFEA performs almost equally to, or occasionally better than, the reference human coders. An important exception is the study by Dupré et al. [21], where human coders outperformed all eight of the AEFEA systems, including FaceReader (version 7), in analyzing both posed and spontaneous EFEs.

Electromyography (EMG)/AEFEA comparison studies found that the two types of systems yielded highly correlated output data; however, compared with EMG, AEFEA systems are much easier to use [25,38,40].

In the case of human rating, test–retest reliability is perceived as an important methodological issue concerning EFEs as well as other aspects of human behavior. However, we found no study exploring the test–retest reliability of any AEFEA system. Most research reports seem to assume, either explicitly or implicitly, that AEFEA systems are based on fully deterministic algorithms, thus taking their perfect test–retest reliability for granted and implying that the stability of their input–output mappings is not an interesting question for research [41].

To fill this gap in our knowledge on the characteristics of AEFEA systems, the present study aimed to explore the test–retest reliability of a specific AEFEA engine, FaceReader 8.0 (henceforth referred to as FR8).

We explored the test–retest reliability of FR8 in a group of subjects with atypical neurocognitive development, namely children with ASD, in addition to a sample of neurotypical children. The quality and quantity of EFEs have diagnostic significance in several conditions, including ASD [9,42]. This makes AEFEA technology an important candidate

research tool in studying these groups and in developing technology-aided screening and diagnostic systems.

In studying EFEs in atypical populations (including both individuals with developmental disorders and those with other psychiatric conditions), three aspects of emotional display seem especially relevant. These may potentially influence the validity and/or reliability of EFE recognition methodologies (both human and automated ratings) in these groups. The three aspects are as follows: (1) the congruence of facial emotions, i.e., the adequacy of an EFE in a specific context [9,43]; (2) the match between the emotion subjectively experienced and the corresponding facial expression [44]; (3) facial morphology [45,46].

Autism spectrum disorders are neurocognitive developmental conditions, and atypical communication of emotions is among their diagnostic markers [47]. Although findings on EFEs in ASD are diverse, studies suggest that all three anomalies mentioned are present in ASD: atypical craniofacial features and development [48–52]; incongruent emotions and mismatch between the expressed and experienced emotions [53–56]. It was also found that some individuals with ASD tend to use EFEs intentionally instead of spontaneously in social contexts, in order to "camouflage" their condition [57]. Therefore, individuals with ASD form a candidate group for exploring whether the test–retest reliability of AEFEA systems is influenced by atypical emotional expression patterns accompanied by atypical craniofacial features.

The present study is part of a research-and-development project with the long-term aim of developing and validating the concept and prototype of a social serious-game-based digital system for the screening of high-functioning cases of ASD among children at kindergarten age [58–61]. A partial prototype of the screening system is under validation and further development; the data analyzed in the present study were collected with this prototype. It records three kinds of behavioral data from the player while they play the game: mouse position and action, gaze focus position, and EFE data (estimates of displayed emotional content, provided by AEFEA on the basis of a video recording of the player's face). The intended output of the system is a three-level risk estimation, based on a combination of the three kinds of collected data. The present study focuses on EFE data.

## 2. Materials and Methods

### 2.1. Study Aims

Given the exploratory nature of the study, we formulated research questions—not specific hypotheses—as follows:

1.  What level and pattern of test–retest reliability does a specific version of AEFEA technology, FR8, provide in a group of typically developing children of kindergarten age, and in a matched group of children with autism spectrum disorder (ASD)?
2.  Does test–retest reliability, if found imperfect, influence the overall pattern of results, especially between-group and between-emotion differences in detected EFE intensities? If so, how?

### 2.2. Participants

A total of 60 children, either with neurotypical development (in this paper, we use the terms "neurotypical" and "typically developing" interchangeably, referring to individuals (children) without any known neurological or psychiatric conditions) (*n* = 31) or with an ASD diagnosis (*n* = 29), were recruited in Hungary to participate in the study. They were recruited via their parents, directly (explicitly), either into the neurotypical (NT) or into the ASD (autism) group, using web advertisements (email circulars and social media). They were all white/Caucasian and without facial artifacts (glasses, etc.). Before collecting EFE data from them, all applicants went through a clinical and psychometric assessment in order to ensure their correct placement into the ASD vs. NT groups and their intellectual abilities, being in the IQ > 85 region, and to collect clinical, demographic, and other background data. Autism-related symptoms were first assessed by the SCQ (Social Communication Questionnaire) [62], and then by the ADOS (Autism Diagnostic

Observation Schedule) [63] and the ADI-R (Autism Diagnostic Interview—Revised) [64]. The latter two are internationally accepted standard tools for autism diagnosis. Intelligence was assessed by the Leiter-R Brief scale [65] and the level of receptive language (grammar) was assessed by the TROG-H (Test for the Reception of Grammar) [66]—the Hungarian adaptation [67]. As the recording of facial expressions was accompanied by eye tracking (although these data were not analyzed in the present study), a questionnaire was used to collect information on any potential eye or visual anomaly. General inclusion criteria comprised the following: age between 36 and 72 months, unimpaired intelligence, and receptive language (grammar). Further inclusion criteria for the NT group comprised the following: SCQ total score < 4; all ADOS and ADI-R scores below the diagnostic cutoff values. Further inclusion criteria for the ASD group comprised the following: all ADOS and ADI-R scores above the diagnostic cutoff values; a clinical diagnosis of ASD from an established diagnostic institution. Exclusion criteria were ophthalmological and neurological conditions interfering with eye tracking; any neurodevelopmental condition, except ASD in the ASD group; significant delay in language, intellectual, or motor development; being on medication influencing the central nervous system; first language being a language other than Hungarian; and any acute or chronic medical condition which might influence the relevant behaviors.

From the total of 97 children who entered the recruitment process, 23 were excluded for not meeting some of the inclusion criteria above or meeting some of the exclusion criteria. Of the 74 children who went on to the EFE-data-collection phase, 14 were excluded from further analysis, due to either failure to collect enough data from them for analysis, or for any emerging doubt concerning the correctness of their diagnostic status (e.g., close to or above the cutoff score in ADOS or ADI-R).

The final sample thus comprised 60 children, 31 being (neurocognitively) typically developing and 29 having an ASD diagnosis. The two groups are well-matched with respect to IQ, receptive grammar, and their performance in the game (as indicated by number of correct (overt) mouse responses). In line with the diagnostic difference between the groups, they showed a significant difference and no overlap in their SCQ scores. There emerged a slightly significant difference in age, with small–moderate effect size; the mean age of the ASD group being higher than that of the NT group. Table 1 shows their key demographic and psychometric characteristics, and matching statistics.

**Table 1.** Demographic and psychometric characteristics of subjects. At the results of statistical difference between groups "ns" means non-significant.

| | NT Group *n* = 31 18 Male/13 Female | | ASD Group *n* = 29 21 Male/8 Female | | Statistical Difference between Groups |
|---|---|---|---|---|---|
| | **Mean (SD)** | **Range** | **Mean (SD)** | **Range** | |
| Age (months) | 53.2 (8.65) | 38–68 | 57.8 (8.65) | 41–70 | t(58) = −2.042; $p$ = 0.046; Cohen's d = 0.26 |
| IQ by Leiter-R Brief | 120.5 (14.04) | 98–147 | 116.3 (13.9) | 91–139 | t(58) = 1.156; $p$ = 0.252 (ns) |
| TROG-H score (receptive grammar) | 118.2 (14.40) | 83–147 | 111.3 (14.93) | 82–153 | t(58) = 1.156; $p$ = 0.074 (ns); Cohen's d = 0.23 |
| | **Median** | **Range** | **Median** | **Range** | |
| SCQ score (ASD symptom severity estimate) | 1 | 0–3 | 21 | 8–29 | Mann–Whitney's U < 0.0001; z = −6.732; $p$ < 0.001 (Mann–Whitney); effect size (stochastic difference; [68]): 1 (large) |
| Game total score (number of correct mouse responses out of the total 24) | 24 | 19–24 | 23 | 2–24 | Mann–Whitney's U = 342; z = −1.660; $p$ = 0.097 (ns); effect size (stochastic difference [68]: 0.239 (large) |

### 2.3. Stimuli

EFE data were collected while children played with the prototype of the computer game. The game content was designed on the basis of empirical studies that revealed

differences in EFEs, visual scanning patterns, or overt intended behavioral responses between neurotypical children and children with ASD of the relevant age. The main theme of the game was based on the story scripts used in an experimental developmental psychology study by Sodian and Frith [69], which compared neurotypical vs. autistic children's ability to understand agents' cooperative or competitive intentions, and to manipulate their behaviors by means of sabotage and deception. Accordingly, our game script centered around controlling the behaviors of social agents (cartoon figures) by sabotage vs. deception, in cooperative and competitive contexts. It consisted of eight such scenes, plus an opening scene to elicit potential group-specific attentional biases, two instruction scenes, and a closing scene.

The game presented predominantly visual stimuli, accompanied occasionally by sounds, such as bird tweets, characters' speech, and various physical noises related to the events shown. Table 2 outlines the scene structure of the game. This section may be divided by subheadings. It should provide a concise and precise description of the experimental results, their interpretation, and the experimental conclusions that can be drawn.

**Table 2.** Outline of the scene structure of the game (adapted from Gyori et al. [61]).

| Scene Theme | Scene Function | Stimuli Presented (Visually) |
|---|---|---|
| "Perceptual preferences" | To evoke spontaneous emotional and gaze responses | Pinwheel, two birds, human agent (narrator) |
| Introduction and instructions—1 | To familiarize with characters, task, controls | |
| Sabotage in cooperative context | | |
| Sabotage in competitive context | To evoke spontaneous emotional and gaze responses, intentional behavioral (mouse) responses | |
| Sabotage in cooperative context | | Human-like characters (narrator, competitor, cooperator), chest, candy, bowl, controls |
| Sabotage in competitive context | | |
| Introduction and instructions—2 | To familiarize with task and controls | |
| Deception in cooperative context | | |
| Deception in competitive context | To evoke spontaneous emotional and gaze responses, intentional behavioral (mouse) responses | |
| Deception in cooperative context | | |
| Deception in competitive context | | |
| Closing | To close the session | Human agent (narrator) |

As progress in a game session depended on some of the player's responses, total play time varied across individuals. Longer response latency resulted in longer game time, and so did erroneous responses, as these resulted in repeating the given trial (up to 3 expositions). The mean total play time was 1482.123 s (SD 88.009, range 1360.933–1688.733). In the ASD group, it was 1502.442 s (SD 102.453, range 1367.233–1688.733); in the NT group it was 1464.426 s (SD 68.387, range 1360.933–1606.933). (Means were calculated only on those subjects who played to the end of the game). The two group means did not differ significantly (independent samples *t*-test, 2-tailed, with different variations).

### 2.4. Setting and Equipment

Recording EFE data from the game took place individually, in standard lab circumstances (lighting held constant, environmental noise reduced to minimal), in the presence of the subject and the experimenter. Parents/caretakers were offered the choice to stay in the lab or to follow the measurement from the neighboring room via a one-way mirror; all chose the latter. Subjects were sitting at a desk in a comfortable fixed chair with adjustable height. Initial viewing distance to the monitor was controlled according to the feedback from the eye-tracking system and was kept at approximately 72 cm. Head and body movements were not constrained, but children were asked to remain seated, and they all did so.

The game was controlled by using the Unity game engine (Unity Technologies) and was running on a desktop-mounted eye-tracking PC (EyeFollower 2, by LC Technologies;

Dresden, Germany). Visual elements of the game were presented on a 24-inch LCD monitor (Benq V2410 ECO 24, Costa Mesa, CA, USA); sounds via a pair of commercial computer loudspeakers. Children could control the game by a computer mouse, designed specifically for young children. A webcam (Logitech®, Newark, CA, USA, C600, 2 MP, 1280 × 720 pixels, max. 30 fps) mounted below the PC monitor made video recordings of the player's face. For 52 subjects, the recording rate was 15 fps (with negligible variability), the video image resolution was 1280 × 720. For the remaining 8 subjects, the recording rate was somewhat higher (between 25.00 and 29.97 fps), and the resolution was lower (between 720 × 480 and 800 × 504). To gain AEFEA data, these recordings were analyzed offline by FR8. These analyses were run on a Lenovo ThinkStation P320, Beijing, China, desktop PC, with key parameters exceeding the configuration recommended by Noldus for running FR8 (in our case: XEON E3 1240v6 3.7 Ghz processor, 16 GB RAM, 256 GB SSD Quadro P600 hard drive) and exceeding the minimum required configuration.

Data processing and analysis were performed using custom-made algorithms in JAVA (Oracle Technology Network, Redwood Shores, CA, USA) and SPSS v25 and v26 (IBM Inc., Armonk, NY, USA).

### 2.5. Procedure

Data collection took place in 3 sessions, individually, for each subject. At the beginning of session 1, children and their parents received detailed information on the purpose and procedure of the study and were asked for their explicit consent. After they provided it, children played with a simple computer game to practice using the one-button computer mouse. After reaching a pre-set criterion, they received brief instruction and then played with the stimulus game, while facial expression data (facial videos) were recorded. In the remaining part of session 1 and in session 2, clinical and psychometric data were collected from the children (Leiter-R, TROG-H, ADOS). In session 3, the ADI-R interview was administered with the parents/caretakers. All these measurements were administered individually by a psychologist trained in the use of these psychometric tools.

### 2.6. Data Processing and Analysis

The input for AEFEA and further analyses were 60 facial video recordings, with a mean length of 1455.862 s (SD = 167.831 s; range: 508.535–1688.733 s) (the mean length of the video recordings was somewhat lower than that of total play times presented above (and the range and SD were greater); this was due to one video recording stopping earlier than the end of the game). These were analyzed by FR8 for gaining EFE intensity data. No preprocessing was performed on the recordings before FR8 analyses. FR8 provides expression intensity values between 0 and 1 on 8 emotions: neutral, happy, sad, angry, surprised, scared, disgusted, and (optionally) contempt. FR8 attempts to assign such intensity values to each frame of the video recording. If it is unable to find the face in any given frame, it logs a "FIND_FAILED" entry for the frame, for each emotion. If it is able to find the face, but unable to fit the face model onto it, it creates a "FIT_FAILED" entry for all emotions for the frame. These data were exported with their time stamps and served as raw data for further analyses.

FR8 provides several settings options to customize EFE analysis. Appendix A Table A1 presents and explains the settings used in this study.

For test–retest reliability analysis, each recorded facial video was analyzed by FR8, repeatedly, a maximum of three times. If the first two subsequent analyses yielded exactly the same EFE intensity values, then no further analysis was run on that specific recording. If, however, no complete agreement was found, then a third analysis was performed.

This repeated analysis was performed by repeatedly sweeping along the set of video recordings in the same order. That is, firstly, all recordings were analyzed once, from subject #1's recording to subject #60's recording, and then the second analysis was carried out in the same order. After checking the level of match between corresponding individual data series, only those recordings were analyzed again (in the same order) where total test–retest

identity was not reached with the second sweep. Those with perfect test–retest identity were omitted from reanalysis. One dataset (file) was gained from each analysis of each recording, and these datasets served as input for further analyses on test–retest reliability. See Results, Section 3, for descriptive data on the repeated analyses and their outcomes.

## 3. Results

### 3.1. Test–Retest Reliability

3.1.1. Test–Retest Differences between First and Second Test Data Series

As a first approach to analyzing test–retest reliability, we compared data pairwise from the first ($test_1$) and second ($test_2$) EFE FR8 analyses of the same videorecording for each subject. Since the recordings differed in length, the number of data points generated by EFE analyses also differed between subjects. For any given subject, we had < length in seconds > times < frames/second > times < emotions: 8 > data points for both $test_1$ and $test_2$. In contrast with the widespread belief reflected by the literature that algorithmic AEFEA has total test–retest reliability, in 12 of 60 of our cases (subjects), the data series gained from $test_1$ and $test_2$ were nonidentical. In these cases, test–retest differences affected a considerable proportion of data points. We calculated the ratio of differing data points between $test_1$ and $test_2$ outputs for each subject; Figure 1 shows the resulting distribution.

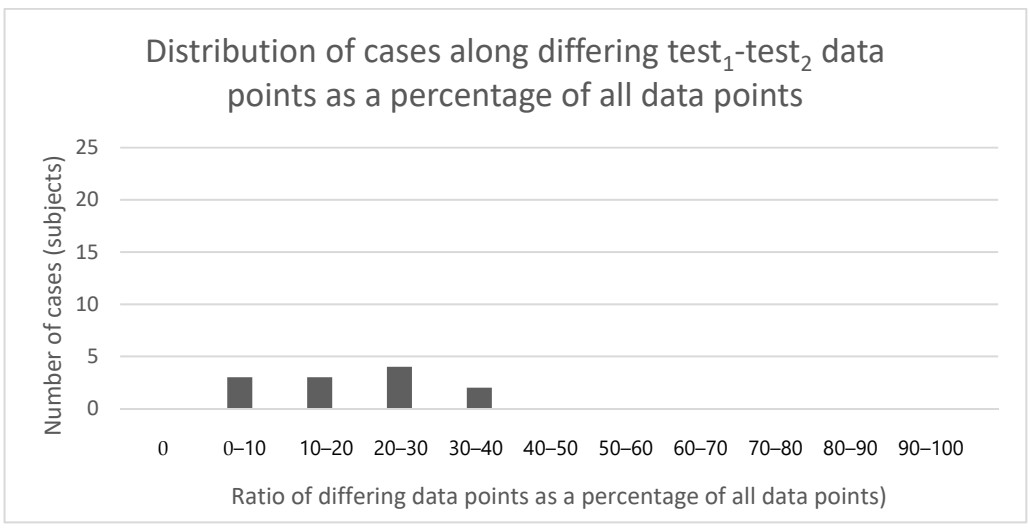

**Figure 1.** Histogram on the 12 subjects with nonidentical data series when comparing $test_1$ and $test_2$ datasets, along differing data points, as a percentage of all data points.

For exploratory purposes, we calculated the mean differences for each subject and emotion, based on individual $test_1$–$test_2$ difference distributions, and then calculated group-level descriptive statistics on each set of individual means; see Appendix A Table A2. Overall, the means of the individual $test_1$–$test_2$ differences were very small, on the order of magnitude of $10^{-3}$–$10^{-7}$ for the eight emotions. The SD values were on a similar order of magnitude—$10^{-3}$–$10^{-6}$.

To explore the individual patterns of the magnitudes of differences between the $test_1$ and $test_2$ output data series, for each subject with nonidentical $test_1$–$test_2$ datasets and for each emotion, we calculated the individual distributions of these different magnitudes. A total of 12 (subjects) $\times$ 8 (emotions) = 96 frequency distributions were thus obtained. For illustration, such difference distributions are shown for two subjects in Appendix A Tables A3 and A4. In these two cases, similarly to other subjects showing $test_1$-$test_2$ differences, the means of the individual emotion difference distributions were low.

For the 12 subjects with nonidentical $test_1$–$test_2$ results, we checked the number of FIND_FAILED (for the failure of finding the face on a video frame) and FIT_FAILED (for the failure of fitting the face model onto the face on a video frame) error signals in the two

subsequent test runs. Importantly, these error signals—and consequently the number of valid data points—are always fully correlated across the eight emotion dimensions in any dataset.

On the subsequent runs (tests), these error indices were almost totally stable for each of the 12 subjects. The differences in their absolute frequencies were not above 3 in any of the 12 cases (cf. the mean total number of data points = 26,139). The FIND_FAILED index was more stable than the FIT_FAILED index (showing difference only in one case in the twelve-case cohort, and only in one data point). This clearly shows that imperfect test–retest reliability did not arise from different distributions of these error signals in the corresponding datasets.

Clearly, these error signals were perfectly stable for each of the 48 subjects with identical test$_1$–test$_2$ datasets.

### 3.1.2. Test–Retest Differences between Second and Third Test Data Series

After finding differences between test$_1$ and test$_2$ data series in one-fifth of the cases in our sample, we explored whether better or perfect test–retest reliability can be reached by further analyses in these cases. This was motivated partly by the fact that, for the majority of the cases, the test$_1$ and test$_2$ datasets were identical. Following the procedure described in the Section 2, we ran another EFE analysis (test$_3$) on each of the 12 recordings and conducted test$_2$–test$_3$ comparisons. From the 12 cases, 5 (3 in the ASD, and 2 in the NT group) showed identical test$_2$ and test$_3$ datasets—these were "stabilized" for test$_3$. In 7 cases, test$_2$–test$_3$ comparisons still revealed differences. Table 3 summarizes the case flow for repeated AEFEAs.

**Table 3.** Case flow for subsequent tests (FR8 analyses) and the number of identical test$_{n-1}$–test$_n$ data series.

|  | ASD Group | NT Group | Total |
|---|:---:|:---:|:---:|
| Number of input cases for test$_1$ and test$_2$/number of identical test$_1$–test$_2$ data series | 29/23 | 31/25 | 60/48 |
| Number of input cases for test$_3$/number of identical test$_2$–test$_3$ data series | 6/3 | 6/2 | 12/5 |
| Remaining "non-stabilized" data series after test$_3$ | 3 | 4 | 7 |

Of the 7 individual dataset pairs with remaining differences in test$_2$–test$_3$ comparisons, 6 showed the same ratios of differing data points in test$_2$–test$_3$ as in test$_1$–test$_2$ contrasts. In one case (subject #65; see Appendix A Tables A4 and A5) there were *more* mismatching data points in the test$_2$–test$_3$ comparison than in the test$_1$–test$_2$ comparison. The absolute values of the test–retest differences between the corresponding data points were again low, in the $[-0.1; 0.1]$ range. All the 7 (subjects) $\times$ 8 (emotions) residual difference means remained below 0.001. The final distribution of percentages of differing data points in test$_2$–test$_3$ contrasts is shown in Figure 2 (analogously to the test$_1$–test$_2$ differences previously shown in Figure 1).

We explored the residual differences for these 7 subjects; see data for individuals in Appendix A Table A6. All the remaining differences are small, with the single highest value being 0.069 (subject #94, *Happy*). Although this is close to a 7% difference on a 0–1 scale, this is a single outlier difference value in the total dataset including data points in the order of magnitude of roughly 11.5 million (i.e., 72 contrasts $\times$ ~20,000 intensity data points $\times$ 8 emotions).

By observing various characteristics of the 7 cases, we attempted to identify any factor that could potentially underlie the lack of "stabilization" (such as group, age, gender, position in the list of data series, video sampling rate, etc.). We failed to identify any such potential factor. In addition, we ran group comparisons along several of these case variables, comparing subject groups corresponding to their stabilized vs. non-stabilized data series. Neither comparing a group of 48 cases to another group of 12 cases (with identical vs. nonidentical data series pairs in test$_1$–test$_2$), nor comparing a 53-case group

to a 7-case group (identical vs. nonidentical data series pairs in test$_2$–test$_3$) revealed any significant cross-group differences. Additionally, as shown in Table 3, the two groups were very balanced in terms of ratios of identical/nonidentical data series pairs.

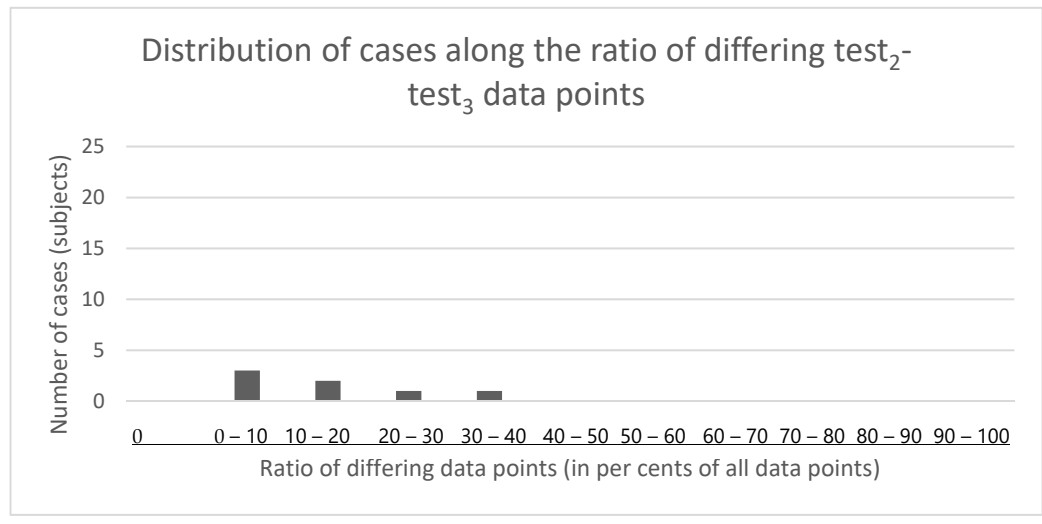

**Figure 2.** Histogram of the 7 subjects with nonidentical data series when comparing their test$_2$ and test$_3$ data, along the ratio of differing data points. (Data of 5 subjects with identical test$_2$–test$_3$ datasets are not included).

### 3.2. Data Robustness (Completeness)

As a background for examining emotion intensities, we briefly explored data robustness (completeness) on the "stabilized" (test$_{stable}$) data series ($n$ = 53), together with the 7 non-stabilized data series from test$_3$. Data robustness or completeness in the present context means the ratio of successfully extracted emotion data within the total data series, as a specific (simple) measure of data quality. As explained above, at any data point, the FR8 system's failure to assigning emotion intensity values may be either due to the failure to find the face in a video frame (FIND_FAILED logged for all emotions), or due to the failure to fit a face model onto the face found (FIT_FAILED logged for all emotions). Thus, data robustness, as a proportion of numerical emotion intensity data logged within all data points (all frames), can be assessed equivalently from any single emotion dimension.

The mean data robustness ratio for all subjects was 0.861 (SD: 0.134; min: 0.436; max: 0.998). The average frequency of face finding failure was 0.0566 (SD: 0.065; min: 0.0001; max: 0.240); the average frequency of face model fitting failure was 0.082 (SD: 0.084; min: 0.002; max: 0.417). This level of data completeness is fairly good, as compared with other recent AEFEA studies [12]. As noted in Section 3.1.1, FIT_FAILED and FIND_FAILED data points remained remarkably stable across tests and comparisons. Importantly for further analyses below, data robustness did not differ between the groups (two-tailed independent samples t-test with different variations; ns).

### 3.3. Between-Emotion and Between-Group Differences in Emotion Intensities

Finally, we explored the pattern of differences in EFE intensities across the eight emotions and between the two groups and examined whether the not-perfect test–retest reliability of AEFEA using FR8 influences this pattern. First, we calculated the emotion intensity means for each subject and each emotion dimension, using their individual data series from their last analysis (test). Accordingly, $n$ = 53 of included individual data series showed total test–retest identity (test$_{stable}$), while $n$ = 7 individual data series came from test$_3$, still not showing total identity to test$_2$ data. Figure 3 shows these means for the groups.

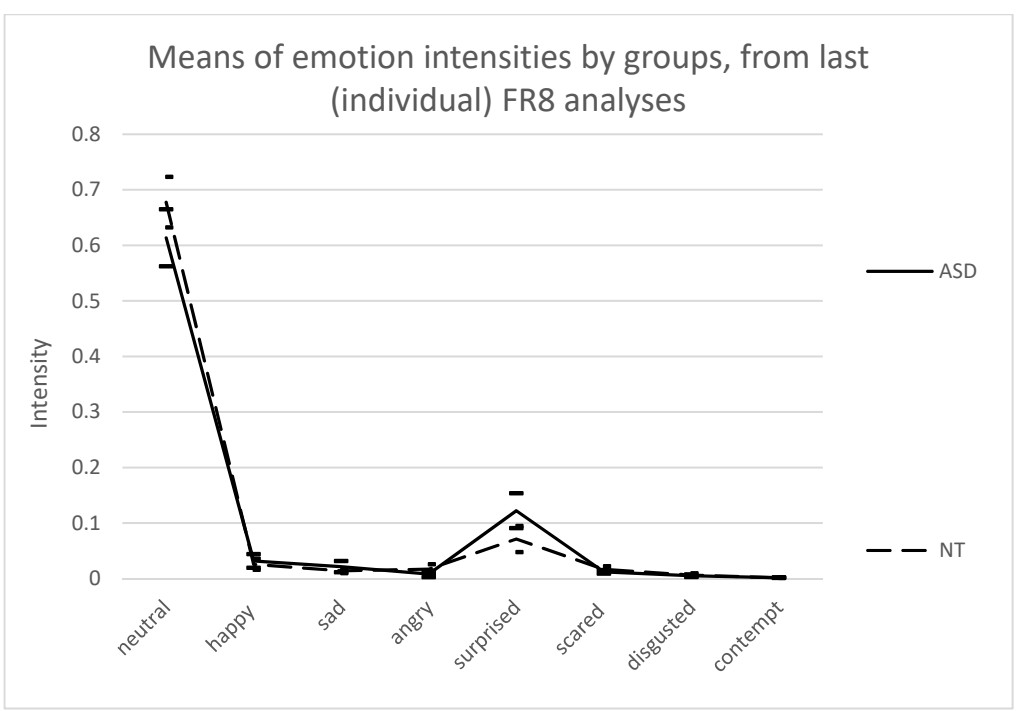

**Figure 3.** Group means of the intensities in the 8 emotions, by group (on data from test$_{stable}$ and remaining nonstable test$_3$ data series, *n* = 60). Error marks: 95% CI.

A two-way mixed ANOVA (two groups × eight emotions; repeated measures) returned a main effect of emotion (Geisser–Greenhouse: F(1.566, 90.801) = 826.832; *p* < 0.001; $\eta^2_{partial}$ = 0.934, $\omega^2$ = 0.930). There was no main effect of diagnosis; the interaction was marginally significant (Geisser–Greenhouse: F(1.00, 58.00) = 3.684, *p* = 0.0599, $\eta^2_{partial}$ = 0.060, $\omega^2$ = 0.042). Repeating the same analysis without data on *Neutral* left the main effect of emotion significant (Geisser–Greenhouse: F(1.665, 96.589) = 51.492; *p* < 0.001, $\eta^2_{partial}$ = 0.470, $\omega^2$ = 0.428); there emerged a main effect of diagnosis—as the equalizing effect of *Neutral* was suppressed, the remaining emotions showed a different overall level in the two groups, with a modest effect size (Welch's test: F(1, 55.4) = 5.055, *p* = 0.0286, $\eta^2_{partial}$ = 0.081, $\omega^2$ = 0.064). The diagnosis–emotion interaction was also significant, but the effect size was small (Geisser–Greenhouse: F(1.665, 96.589) = 4.456; *p* = 0.0391; $\eta^2_{partial}$ = 0.071, $\omega^2$ = 0.049). When *Surprised* was also removed, the remaining six emotions still produced a significant main effect of emotion (Geisser–Greenhouse: F(2.525, 146.452) = 14.98; *p* < 0.001; $\eta^2_{partial}$ = 0.205, $\omega^2$ = 0.165) but neither a main effect of diagnosis nor an interaction remained. The main effect of emotion in this case was due to a number of between-level differences, as indicated by Tukey's test (*Happy* > [*Angry*, *Scared*, *Disgusted*, *Contempt*]; *Sad* > [*Disgusted*, *Contempt*]; *Scared* > *Contempt*; only those differences are listed for which *p* < 0.01). However, all six of these emotion means were very low in both groups (total group means: ASD: 0.0133; NT: 0.0135). The level of *Neutral* differed only marginally significantly between groups (W(56.6) = −1.751; *p* = 0.0854; Cohen's d: −0.454; mean difference: −0.062, CI95: [−0.131; 0.009]); while the level of *Surprised* differed significantly in the two groups (Welch test: W(52.8) = 2.36; *p* = 0.022; Cohen's d = 0.615, mean difference: 0.0469; CI95: [0.0071, 0.0876]); however, these differences would obviously not survive a type I error correction for eight emotions.

To explore if the differences between the initial (test$_1$) dataset and the final dataset influence the results above, the same ANOVA was repeated using the test$_1$ data of each subject. All results were very similar, so much so that the overall emotion intensity means plot was not noticeably different from Figure 3—therefore, we will not present it here.

The two-way mixed ANOVA (2 groups X 8 emotions) returned a main effect of emotion, here, too (Geisser–Greenhouse: F(1.0, 58.0) = 827.835; *p* < 0.001; $\eta^2_{partial}$ = 0.931, $\omega^2$ = 0.935).

There was no main effect of diagnosis; the interaction was, again, marginally significant (Geisser–Greenhouse: $F(1.00, 58.00) = 3.664$, $p = 0.0605$, $\eta^2_{partial} = 0.059$, $\omega^2 = 0.041$). Repeating the same analysis without data on *Neutral* left the main effect of emotion significant (Geisser–Greenhouse: $F(1.0, 58.0) = 51.387$; $p < 0.001$, $\eta^2_{partial} = 0.470$, $\omega^2 = 0.427$), and there emerged a main effect of diagnosis; as the equalizing effect of *Neutral* was, again, suppressed, the remaining emotions showed a different pattern in the two groups, with a modest effect size (Welch's test: $F(1; 55.4) = 5.011$, $p = 0.029$, $\eta^2_{partial} = 0.080$, $\omega^2 = 0.063$). The interaction was again significant, with a small effect size (Geisser–Greenhouse: $F(1.0, 58.0) = 4.416$; $p = 0.0400$; $\eta^2_{partial} = 0.071$, $\omega^2 = 0.049$). When *Surprised* was also removed, the remaining six emotions still produced a significant main effect of emotion (Geisser–Greenhouse: $F(1.0, 58.0) = 14.955$; $p < 0.001$; $\eta^2_{partial} = 0.205$, $\omega^2 = 0.165$) but neither a main effect of diagnosis nor an interaction remained. The post hoc comparisons between levels of emotion returned the following differences at $p < 0.01$: *Happy* > [*Angry, Sad, Disgusted, Contempt*]; *Sad* > [*Disgusted, Contempt*]; *Scared* > *Contempt*. Once again, all six of these emotion means were at very low intensities in both groups (total group means: ASD: 0.0133; NT: 0.0135). The level of *Neutral* differed, again, only marginally significantly between groups (Welch test: $d(56.6) = -1.748$; $p = 0.0859$; Cohen's d: $-0.453$; mean difference: $-0.061$; CI95: [$-0.131$; 0.009]); the level of *Surprised* differed significantly in the two groups (Welch test: $d(52.9) = 2.350$; $p = 0.0226$; Cohen's d = 0.612, mean difference: 0.0469; CI95: [0.0069, 0.0873].

In sum, $test_1$ vs. the "final" differences in data series did not lead to any significant difference in the pattern of overall emotion intensity means among the different emotions and the groups.

## 4. Discussion

Our most important finding is that the test–retest reliability of a specific version of AEFEA technology, the FR8, is very high; however, it is not perfect. Having analyzed the same set of facial video recordings by FR8 twice ($test_1$ and $test_2$), we found that in 12 of the 60 cases, $test_1$–$test_2$ differences arose. These differences appeared in all eight of the emotion dimensions, in up to 40% of the data points altogether, but were very small in their absolute values. This suggests a remarkably high test–retest reliability as compared with human coders [14], but lags behind a *perfect* test–retest reliability, which is often explicitly or implicitly expected from algorithmic AEFEA technologies [41].

Another set of analyses ($test_3$) of those facial video recordings which did not initially show total $test_1$–$test_2$ identity led to further test–retest convergence or stabilization in five cases. In the seven remaining cases, however, there were both $test_1$–$test_2$ and $test_2$–$test_3$ differences between the individual data series.

From the perspective of utilizing AEFEA technology in screening, diagnosis, and other health technologies, it is of special importance that our analyses did not reveal any cross-group difference in test–retest reliability. Data series from neurotypical children and from children with ASD were very similarly prone to imperfect test–retest reliability and showed very similar stabilization tendency in proportional cases. Additionally, a similar proportion of cases in both groups was left non-stabilized after three analyses. The pattern we found in the cross-group and cross-emotion differences in emotion intensities fits well with the series of previous findings on emotional display in ASD [53,56,70]. To the best of our knowledge, the test–retest reliability of AEFEA techniques has not been explored so far; therefore, these findings are unprecedented in the literature.

We address four questions: (1) Is it possible that the findings on imperfect test–retest reliability are artefacts? (2) If they are not artefacts, what may cause the imperfect test–retest reliability? (3) What factor(s) may determine which cases or data series are affected by it? (4) Does imperfect test–retest reliability influence the fundamental results?

In our view, it is crucial to consider the issue of whether the above findings may be artefacts. We see two factors that could potentially lead to the emergence of spurious results. Firstly, non-sufficient hardware performance in the course of FR8 analyses might—as pilot

analyses we ran suggested—lead to noise in the emotion intensity data that could deteriorate test–retest reliability. However, the hardware configuration we used considerably surpassed that recommended by Noldus for using FR8. In addition, the partial test–retest convergence of subsequent data series could not be explained by such a factor.

Software updates of FR8, arriving between subsequent analyses and changing the input/output mapping of the system, represent another candidate. These could explain both test–retest differences and their later partial decrease (assuming subsequent updates converge towards a stable mapping). Updates, however, were turned off for FR8 during and between analyses. A "software update effect", moreover, could not explain why some data series converged in subsequent analyses while some failed to stabilize within two subsequent comparisons.

Finally, we note that data completeness in our data series was fairly good, as compared with other recent studies [12]. Even if it were lower than usual, it would be surprising and methodologically significant if the sparsity of data could lead to imperfect test–retest reliability.

The following question arises: which feature of FR8 explains its imperfect test–retest reliability? In this study, we treated the FR8 system as a black box. We refrained from reverse-engineering it and from speculating about possible specific causes. Nevertheless, we briefly raise three possibilities. One is that artificial neural network (ANN) technology, which is a feature of FR8, may underlie our findings. ANNs "naturally" respond differently to repeated stimuli in the learning phase, and they also tend to converge onto more stable input/output mappings. In the case of such an ANN learning effect, the methodological consequences of imperfect initial test–retest reliability are still to be considered. Secondly, the "continuous calibration" feature of FaceReader arises as a candidate for changing analysis characteristics. In our understanding, continuous calibration does modulate emotion estimation output: it adjusts facial intensity values in an ongoing analysis continuously, as a function of previous intensity values in that analysis. However, FaceReader uses a fixed formula for this process—that is, if the detected raw intensities are stable across analyses, then continuous calibration in itself could not add "noise" to these. Thirdly, another, more "profane" possibility—suggested by our informal experimentation with this factor—is that hardware requirements for running FR8 might be underestimated by its producer, and insufficient hardware resources lead to unstable performance. However, this explanation would go against the fact that most data series showed either test–retest stability or stabilizing tendency across the subsequent re-analyses.

In this study, 20% of our cases/initial data series were affected by nonperfect test–retest reliability. Empirical research using AEFEA could benefit from understanding which cases/data series are more prone to this than others. Although we examined several candidate subject- or data-related factors, we failed to identify any that could determine which subsets of cases showed perfect/imperfect test–retest reliability.

The question of whether imperfect test–retest reliability of AEFEA technology substantially influences results is of crucial importance in a field of research where this issue has been ignored so far. We analyzed the patterns of cross-emotion and cross-group differences, both in the first and the last (mostly stabilized) tests' data series, to see whether there were fundamental differences in the two sets of results. The overall patterns of emotion intensity differences remained closely similar. That is, the less-than-perfect test–retest reliability did not substantially influence these results in either group.

Two cautionary notes must be made, however. Here, we used relatively long video recordings as input for AEFEA, resulting in relatively big emotion intensity data series. Other studies have used much shorter recordings, and therefore obtained smaller data series [41,71]. We did not explore the distribution of test–retest differences along the data series, leaving it possible that potential "locally high" test–retest differences, especially in shorter data series, may have a greater effect on the patterns of the results. The overall pattern of emotion intensities itself may also affect the test–retest differences. Data from different emotions may be differently affected by the test–retest differences. Exploring this possibility was beyond the scope of this study.

Finally, we wish to point to the fact that the validity of the data gained repeatedly by FR8 was not addressed or examined in this study. Therefore, we cannot know whether the initial or the final ("stabilized") data series were more valid. (Although small test–retest differences generally suggest little, if any, potential difference in their validity.) Accordingly, we cannot provide any insight into which should be preferred in further analyses and in reaching study conclusions, where the validity of emotion intensity data is essential.

## 5. Conclusions

Our study demonstrated the high but imperfect test–retest reliability of a specific AEFEA system—FR8. As the second main finding, we demonstrated that repeated analyses by FR8 can, in some cases, lead to the "stabilization" of datasets. Thirdly, the test–retest differences did not influence the patterns of cross-emotion and cross-group effects and interactions. Thus, our study does not question the validity of previous results gained by AEFEA technology. Finally, no specific pattern of test–retest reliability emerged in the group of children with ASD.

As a first exploration of the issue, our study leaves several key questions open about generalizing our results across AEFEA systems, subject groups, emotions, and emotion intensity patterns. Therefore, there are questions to be explored: whether imperfect test–retest reliability and the occasional "stabilization" effect are specific to FR8, or whether other AEFEA systems also show them; whether these phenomena are specific to our white/Caucasian child-aged sample(s) or not; and whether they arise on inputs with other emotions and/or emotion intensity patterns significantly different from those used in this study.

Although the test–retest emotion intensity differences were small, and the imperfect test–retest reliability did not have an essential effect on group-level emotion intensity differences, it may have methodological significance when shorter (sections of) data series are analyzed, and when overall intensity patterns of emotions are different from those found in this study.

It also remains to be determined whether first-test data series are less or more valid than subsequent ones, and whether gaining "stabilized" datasets (i.e., reaching perfect test–retest validity by repeated analyses) increases the validity.

Our results do not question the validity of previous results using AEFEA technology. Instead, they suggest that it is necessary to further explore and understand this key characteristic of AEFEA systems.

Finally, we ask the following question: which aspects of the functioning of FR8 account for the high but imperfect test–retest reliability and may be significant for the further development of AEFEA technologies?

**Author Contributions:** Conceptualization, M.G., Z.B. and K.S.; methodology, M.G., Z.B. and K.S.; software, M.G. and Z.J.; validation, M.G., Z.B. and K.S.; formal analysis, M.G., Z.J., Z.B. and K.S.; data curation, M.G., Z.J., Z.B. and B.B.; writing—original draft preparation, M.G., Z.B., Z.J. and K.S.; writing—review and editing, M.G., Z.B., Z.J. and K.S.; visualization, Z.J.; supervision, K.S.; project administration, B.B.; funding acquisition, M.G. All authors have read and agreed to the published version of the manuscript.

**Funding:** Various aspects of the present study and the underlying R&D project have been funded by a grant within the EIT ICT Labs Hungarian Node (PI: András Lőrincz, ELTE Eötvös Loránd University); a TÁMOP grant co-financed by the European Union and the government of Hungary (TÁMOP 4.2.1./B-09/KMR—2010-0003); the ELTE University Thematic Excellence Program 2020, supported by the National Research—Development and Innovation Office—TKP2020-IKA-05; the Social Innovation Laboratory (TINLAB) Project at ELTE University; and a fundraising campaign by the IndaGaléria (Budapest). The first, third, and fifth authors were supported via a research grant from the Hungarian Academy of Sciences, within its Content Pedagogy Research Program (2016–2021 period). The Foundation for the Development of Special Education (GYFA; Budapest) contributed to the funding of the research at several points.

**Institutional Review Board Statement:** The study was conducted in accordance with the Declaration of Helsinki, and approved by the Institutional Research Ethics Committee of the Bárczi Gusztáv

Faculty of Special Needs Education, ELTE Eötvös Loránd University (protocol code KEB/2016/001 and date of approval: 3 February 2016).

**Informed Consent Statement:** Informed consent was obtained from all subjects involved in the study. All participant children were explicitly informed about the aims and circumstances of the measurements and on their right to withdraw from participation at any time, before each session. They gave oral confirmation of their understanding and consent. The caregiver (parent) of each child received both oral and written information about the aims and circumstances of the measurements, the processing and handling of the data gained, as well as on their relevant rights. They gave written consent on their own and their child's behalf. At the beginning of each session, they were reminded of their relevant rights and confirmed their written consent orally.

**Data Availability Statement:** The data presented in this study are openly available at OSF.io at DOI 10.17605/OSF.IO/27EQD. The authors kindly encourage anyone to explore the data but also ask to contact us prior to any reproduction and/or further sharing of the data and/or making public any result from their analysis by any means.

**Acknowledgments:** Authors thank all participating children and their parents/caretakers.

**Conflicts of Interest:** The authors declare no conflict of interest. The funders had no role in the design of the study; in the collection, analyses, or interpretation of data; in the writing of the manuscript, or in the decision to publish the results.

## Appendix A

**Table A1.** Relevant FR8 analysis settings and their rationale.

| Analysis Parameter | Applied Setting | Rationale |
|---|---|---|
| Face model | Children | In accordance with subjects' age |
| Calibration | Continuous calibration | Aims to correct for person-specific biases in facial expressions, thus not requiring an initial input of a neutral face of each subject |
| Classification | Smoothen classification | Raw classification data are smoothened in relation to between-frame time gap (default setting) |
| Image rotation | No | All faces were recorded in upright position |
| Video sample rate | Every frame | No recorded frame to be skipped |
| Contempt | Treat contempt as an emotional state | To gain intensity values for "contempt" |
| Preset for face-finding function | Find all faces (slow) | The most robust setting in terms of size of potential target face on the frame (minimum face fraction = 0.08) |
| Preset for face-modeling function | Maximum accuracy (slow) | Robust setting in terms of the maximum number of iterations per frame to find face (12) |
| Engine | Use deep face engine | Slower but more accurate analysis which attempts to model the face even from partial information |
| Maximum face model error | 0.6 | Measure of error for estimating if face model is valid |
| Size of interest, maximum face fraction | 1 | Maximum value, allows the face surface to be the same as the entire image |
| Size of interest, face size scaling factor | 1.1 | The default factor value for increasing face size within minimum and maximum, for the analysis |

**Table A2.** Descriptive statistics of test$_1$–test$_2$ emotion intensity differences averaged over total play time (all data points), separately for each emotion. Group means were calculated from individual means. Cases with identical test$_1$–test$_2$ data series are not included. Case numbers are too low for normality test. Notation: *: $p < 0.05$; **: $p < 0.01$.

| Group | Emotion | Individual Test$_1$–Test$_2$ Emotion Intensity Differences | | | | | | |
|---|---|---|---|---|---|---|---|---|
| | | **Mean** | **SD** | **Median** | **Minimum** | **Maximum** | **Skewness** | **Kurtosis** |
| ASD ($n = 6$) | Neutral | 0.00123 | 0.003744 | −0.0001 | −0.0009 | 0.0088 | 2.387 * | 5.767 ** |
| | Happy | −0.00089 | 0.001878 | 0 | −0.0047 | 0.0002 | −2.286 * | 5.303 ** |
| | Sad | 0.000103 | 0.000201 | 0 | 0 | 0.0005 | 2.328 * | 5.535 ** |
| | Angry | $-7.3 \times 10^{-5}$ | 0.000197 | 0 | −0.0005 | 0.0001 | −2.280 * | 5.347 ** |
| | Surprised | −0.00322 | 0.007911 | 0 | −0.0194 | 0.0001 | −2.449 * | 6.000 ** |
| | Scared | 0.001027 | 0.002513 | 0 | 0 | 0.0062 | 2.449 * | 5.998 ** |
| | Disgusted | −0.00022 | 0.000389 | −0.0001 | −0.001 | 0.0001 | −2.139 * | 4.839 * |
| | Contempt | $-5.9 \times 10^{-5}$ | 0.000147 | 0 | −0.0004 | 0 | −2.448 * | 5.996 ** |
| NT ($n = 6$) | Neutral | $2.85 \times 10^{-6}$ | 0.000327 | 0 | −0.0004 | 0.0006 | 0.889 | 2.828 |
| | Happy | −0.00017 | 0.000315 | 0 | −0.0008 | 0 | −2.211 * | 4.956 * |
| | Sad | $2.02 \times 10^{-5}$ | $5.2 \times 10^{-5}$ | 0 | 0 | 0.0001 | 0 | 0 |
| | Angry | 0.000255 | 0.000641 | 0 | 0 | 0.0016 | 2.447 * | 5.991 ** |
| | Surprised | $-3.2 \times 10^{-6}$ | $6.15 \times 10^{-5}$ | 0 | −0.0001 | 0.0001 | 0 | 0 |
| | Scared | 0.000264 | 0.000681 | 0 | 0 | 0.0017 | 2.446 * | 5.987 ** |
| | Disgusted | $-2.3 \times 10^{-5}$ | 0.000252 | 0 | −0.0004 | 0.0004 | −0.065 | 2.171 |
| | Contempt | $-5.8 \times 10^{-7}$ | $1.39 \times 10^{-6}$ | 0 | 0 | 0 | 0 | 0 |

**Table A3.** Example distributions of test$_1$–test$_2$ differences, with corresponding means, separately for the 8 emotions: subject #2 (ASD). Test$_2$–test$_3$ data for subject #2 were identical. Frequency distributions include only data points with intensity values assigned in both test$_1$ and test$_2$ (therefore numerical difference could be calculated); however, matching data points with zero difference were omitted. *Nonzero percent* row shows the relative frequencies of mismatching data points for each emotion. As emotion intensities are scaled from 0 to 1 by FR8, the difference between two corresponding data points ranges between −1 and 1; in absolute value between 0 and 1.

| Magnitude Range | Neutral | Happy | Sad | Angry | Surprised | Scared | Disgusted | Contempt |
|---|---|---|---|---|---|---|---|---|
| diff = −1 | 0 | 0 | 0 | 0 | 0 | 0 | 0 | 0 |
| −1 < diff ≤ −0.9 | 0 | 0 | 0 | 0 | 0 | 0 | 0 | 0 |
| −0.9 <diff ≤ −0.8 | 0 | 0 | 0 | 0 | 0 | 0 | 0 | 0 |
| −0.8 < diff ≤ −0.7 | 0 | 0 | 0 | 0 | 0 | 0 | 0 | 0 |
| −0.7 < diff ≤ −0.6 | 0 | 0 | 0 | 0 | 0 | 0 | 0 | 0 |
| −0.6 < diff ≤ −0.5 | 0 | 0 | 0 | 0 | 0 | 0 | 0 | 0 |
| −0.5 < diff ≤ −0.4 | 0 | 0 | 0 | 0 | 0 | 0 | 0 | 0 |
| −0.4 < diff ≤ −0.3 | 0 | 0 | 0 | 0 | 0 | 0 | 0 | 0 |
| −0.3 < diff ≤ −0.2 | 0 | 0 | 0 | 0 | 0 | 0 | 0 | 0 |
| −0.2 < diff ≤ −0.1 | 0 | 0 | 0 | 0 | 168 | 0 | 0 | 0 |
| −0.1 < diff ≤ 0 | 5638 | 7493 | 5920 | 10,405 | 9736 | 397 | 9593 | 9680 |
| 0 < diff ≤ 0.1 | 11,661 | 511 | 9113 | 15 | 5661 | 11,322 | 1676 | 1 |
| 0.1 < diff ≤ 0.2 | 0 | 0 | 0 | 0 | 242 | 0 | 0 | 0 |
| 0.2 < diff ≤ 0.3 | 0 | 0 | 0 | 0 | 0 | 0 | 0 | 0 |
| 0.3 < diff ≤ 0.4 | 0 | 0 | 0 | 0 | 0 | 0 | 0 | 0 |
| 0.4 < diff ≤ 0.5 | 0 | 0 | 0 | 0 | 0 | 0 | 0 | 0 |

**Table A3.** *Cont*.

| Magnitude Range | Neutral | Happy | Sad | Angry | Surprised | Scared | Disgusted | Contempt |
|---|---|---|---|---|---|---|---|---|
| 0.5 < diff ≤ 0.6 | 0 | 0 | 0 | 0 | 0 | 0 | 0 | 0 |
| 0.6 < diff ≤ 0.7 | 0 | 0 | 0 | 0 | 0 | 0 | 0 | 0 |
| 0.7 < diff ≤ 0.8 | 0 | 0 | 0 | 0 | 0 | 0 | 0 | 0 |
| 0.8 < diff ≤ 0.9 | 0 | 0 | 0 | 0 | 0 | 0 | 0 | 0 |
| 0.9 < diff ≤ 1 | 0 | 0 | 0 | 0 | 0 | 0 | 0 | 0 |
| **Mean** | 0.008829 | −0.00466 | 0.000509 | −0.00047 | −0.01937 | 0.006156 | −0.0002 | −0.00036 |
| **Nonzero percent** | 0.395333 | 0.182915 | 0.343549 | 0.238128 | 0.361237 | 0.267814 | 0.25753 | 0.22124 |
| **Total frames** | 43,758 | | | | | | | |

**Table A4.** Example distributions of $test_1$–$test_2$ differences, with corresponding means, separately for the 8 emotions: subject #65 (ASD). Frequency distributions include only data points with intensity values assigned in both $test_1$ and $test_2$ (therefore numerical difference could be calculated); however, matching data points with zero difference were omitted. *Nonzero percent* row shows the relative frequencies of mismatching data points for each emotion. As emotion intensities are scaled from 0 to 1 by FR8, the difference between two corresponding data points ranges between −1 and 1; in absolute value between 0 and 1.

| Magnitude Range | Neutral | Happy | Sad | Angry | Surprised | Scared | Disgusted | Contempt |
|---|---|---|---|---|---|---|---|---|
| diff = −1 | 0 | 0 | 0 | 0 | 0 | 0 | 0 | 0 |
| −1 < diff ≤ −0.9 | 0 | 0 | 0 | 0 | 0 | 0 | 0 | 0 |
| −0.9 < diff ≤ −0.8 | 0 | 0 | 0 | 0 | 0 | 0 | 0 | 0 |
| −0.8 < diff ≤ −0.7 | 0 | 0 | 0 | 0 | 0 | 0 | 0 | 0 |
| −0.7 < diff ≤ −0.6 | 0 | 0 | 0 | 0 | 0 | 0 | 0 | 0 |
| −0.6 < diff ≤ −0.5 | 1 | 0 | 0 | 0 | 0 | 0 | 0 | 0 |
| −0.5 < diff ≤ −0.4 | 0 | 0 | 0 | 0 | 0 | 0 | 0 | 0 |
| −0.4 < diff ≤ −0.3 | 0 | 0 | 0 | 0 | 0 | 0 | 0 | 0 |
| −0.3 < diff ≤ −0.2 | 0 | 0 | 0 | 0 | 0 | 0 | 1 | 0 |
| −0.2 < diff ≤ −0.1 | 0 | 0 | 0 | 0 | 0 | 0 | 0 | 0 |
| −0.1 < diff ≤ 0 | 205 | 1 | 635 | 312 | 479 | 44 | 83 | 516 |
| 0 < diff ≤ 0.1 | 535 | 262 | 0 | 72 | 0 | 281 | 268 | 0 |
| 0.1 < diff ≤ 0.2 | 0 | 0 | 0 | 0 | 0 | 0 | 0 | 0 |
| 0.2 < diff ≤ 0.3 | 0 | 0 | 0 | 0 | 0 | 0 | 0 | 0 |
| 0.3 < diff ≤ 0.4 | 0 | 0 | 0 | 0 | 0 | 0 | 0 | 0 |
| 0.4 < diff ≤ 0.5 | 0 | 0 | 0 | 0 | 0 | 0 | 0 | 0 |
| 0.5 < diff ≤ 0.6 | 0 | 0 | 0 | 0 | 0 | 0 | 0 | 0 |
| 0.6 < diff ≤ 0.7 | 0 | 0 | 0 | 0 | 0 | 0 | 0 | 0 |
| 0.7 < diff ≤ 0.8 | 0 | 0 | 0 | 0 | 0 | 0 | 0 | 0 |
| 0.8 < diff ≤ 0.9 | 0 | 0 | 0 | 0 | 0 | 0 | 0 | 0 |
| 0.9 < diff ≤ 1 | 0 | 0 | 0 | 0 | 0 | 0 | 0 | 0 |
| **Mean** | −0.00092 | $-4.1 \times 10^{-5}$ | $-8.8 \times 10^{-6}$ | $2.23 \times 10^{-5}$ | $-2.5 \times 10^{-6}$ | $-3.2 \times 10^{-5}$ | −0.00099 | $-6.19 \times 10^{-8}$ |
| **Nonzero percent** | 0.031993 | 0.011355 | 0.027417 | 0.01658 | 0.020681 | 0.014032 | 0.015198 | 0.022279 |
| **Total frames** | 23,161 | | | | | | | |

**Table A5.** Distributions of emotion intensity differences for Subject #65 (ASD) from $\text{test}_2$–$\text{test}_3$ comparison. Subject #65 was the only participant whose number of differing data points increased from $\text{test}_1$–$\text{test}_2$ to $\text{test}_2$–$\text{test}_3$ comparison. Still, the corresponding distribution means are quite low in both comparisons (cf. Appendix A Table A4).

| Magnitude Range | Neutral | Happy | Sad | Angry | Surprised | Scared | Disgusted | Contempt |
|---|---|---|---|---|---|---|---|---|
| diff = −1 | 0 | 0 | 0 | 0 | 0 | 0 | 0 | 0 |
| −1 < diff ≤ −0.9 | 0 | 0 | 0 | 0 | 0 | 0 | 0 | 0 |
| −0.9 < diff ≤ −0.8 | 0 | 0 | 0 | 0 | 0 | 0 | 0 | 0 |
| −0.8 < diff ≤ −0.7 | 0 | 0 | 0 | 0 | 0 | 0 | 0 | 0 |
| −0.7 < diff ≤ −0.6 | 0 | 0 | 0 | 0 | 0 | 0 | 0 | 0 |
| −0.6 < diff ≤ −0.5 | 0 | 0 | 0 | 0 | 0 | 0 | 0 | 0 |
| −0.5 < diff ≤ −0.4 | 0 | 0 | 0 | 0 | 0 | 0 | 0 | 0 |
| −0.4 < diff ≤ −0.3 | 0 | 0 | 0 | 0 | 0 | 0 | 0 | 0 |
| −0.3 < diff ≤ −0.2 | 0 | 0 | 0 | 0 | 0 | 0 | 0 | 0 |
| −0.2 < diff ≤ −0.1 | 0 | 0 | 0 | 0 | 0 | 0 | 0 | 0 |
| −0.1 < diff ≤ 0 | 2506 | 561 | 6427 | 139 | 46 | 4813 | 780 | 4570 |
| 0 < diff ≤ 0.1 | 5588 | 1 | 600 | 3043 | 4132 | 85 | 83 | 506 |
| 0.1 < diff ≤ 0.2 | 0 | 0 | 0 | 0 | 0 | 0 | 0 | 0 |
| 0.2 < diff ≤ 0.3 | 0 | 0 | 0 | 0 | 0 | 0 | 1 | 0 |
| 0.3 < diff ≤ 0.4 | 0 | 0 | 0 | 0 | 0 | 0 | 0 | 0 |
| 0.4 < diff ≤ 0.5 | 0 | 0 | 0 | 0 | 0 | 0 | 0 | 0 |
| 0.5 < diff ≤ 0.6 | 1 | 0 | 0 | 0 | 0 | 0 | 0 | 0 |
| 0.6 < diff ≤ 0.7 | 0 | 0 | 0 | 0 | 0 | 0 | 0 | 0 |
| 0.7 < diff < = 0.8 | 0 | 0 | 0 | 0 | 0 | 0 | 0 | 0 |
| 0.8 < diff ≤ 0.9 | 0 | 0 | 0 | 0 | 0 | 0 | 0 | 0 |
| 0.9 < diff ≤ 1 | 0 | 0 | 0 | 0 | 0 | 0 | 0 | 0 |
| **Mean** | $9 \times 10^{-5}$ | $2 \times 10^{-5}$ | $4 \times 10^{-5}$ | $-3 \times 10^{-6}$ | $-5.7 \times 10^{-5}$ | $3 \times 10^{-6}$ | 0.000403 | $-2.29 \times 10^{-8}$ |
| **Nonzero percent** | 0.3495 | 0.0243 | 0.3034 | 0.1374 | 0.180389 | 0.2115 | 0.037304 | 0.2191615 |
| **Total frames:** | 23,161 | | | | | | | |

**Table A6.** Residual differences between $\text{test}_2$ and $\text{test}_3$ intensity data for the 7 cases which did not "stabilize" for the $\text{test}_2$–$\text{test}_3$ comparison. All maxima are absolute values. Frequencies of differing data points are shown relative to (i) the number of all data points: "rel. freq. in all data", and to (ii) the number of "valid" data points with numerical intensity values (thus excluding data points with FIT_FAILED or FIND_FAILED error signals): "rel. freq. in valid data".

| | | Neutral | Happy | Sad | Angry | Surprised | Scared | Disgusted | Contempt |
|---|---|---|---|---|---|---|---|---|---|
| | | \multicolumn | | | | | | | |

| | | Neutral | Happy | Sad | Angry | Surprised | Scared | Disgusted | Contempt |
|---|---|---|---|---|---|---|---|---|---|
| **#27** | Mean | $-2.3 \times 10^{-6}$ | $3.37 \times 10^{-6}$ | $1.2 \times 10^{-6}$ | $7.29 \times 10^{-7}$ | $-2.4 \times 10^{-5}$ | $1.13 \times 10^{-6}$ | $4.27 \times 10^{-5}$ | $4.46 \times 10^{-8}$ |
| | maximum | 0.001467 | 0.001972 | $3.79 \times 10^{-6}$ | 0.000158 | 0.011318 | $2.54 \times 10^{-6}$ | 0.007265 | $1.49 \times 10^{-7}$ |
| | rel. freq. in all data | 0.084117 | 0.027527 | 0.056829 | 0.026808 | 0.08287 | 0.045943 | 0.025993 | 0.032611 |
| | rel. freq. in valid data | 0.10206 | 0.033399 | 0.068951 | 0.032526 | 0.100547 | 0.055743 | 0.031537 | 0.039567 |
| **#43** | Mean | $-6.7 \times 10^{-6}$ | $9.14 \times 10^{-8}$ | $8.03 \times 10^{-6}$ | $-6.7 \times 10^{-7}$ | $-1.1 \times 10^{-6}$ | $-4.1 \times 10^{-7}$ | $-4.6 \times 10^{-8}$ | $-2.4 \times 10^{-8}$ |
| | maximum | 0.005283 | $5.29 \times 10^{-5}$ | 0.002069 | $1.79 \times 10^{-6}$ | $1.39 \times 10^{-5}$ | $9.04 \times 10^{-7}$ | $1.1 \times 10^{-7}$ | $1 \times 10^{-7}$ |
| | rel. freq. in all data | 0.216552 | 0.101549 | 0.109482 | 0.105168 | 0.159352 | 0.175821 | 0.040314 | 0.166868 |
| | rel. freq. in valid data | 0.263016 | 0.123338 | 0.132973 | 0.127733 | 0.193543 | 0.213545 | 0.048963 | 0.202671 |
| **#46** | Mean | $3.72 \times 10^{-6}$ | $-9 \times 10^{-6}$ | $-1.6 \times 10^{-7}$ | $-2.8 \times 10^{-8}$ | $-1.3 \times 10^{-5}$ | $1.31 \times 10^{-7}$ | $-2.2 \times 10^{-8}$ | $1.32 \times 10^{-7}$ |
| | maximum | 0.004441 | 0.003993 | $6 \times 10^{-7}$ | $1.32 \times 10^{-7}$ | 0.003331 | $3.86 \times 10^{-5}$ | $1.67 \times 10^{-6}$ | $5.37 \times 10^{-7}$ |
| | rel. freq. in all data | 0.297395 | 0.215251 | 0.133531 | 0.218219 | 0.261457 | 0.24436 | 0.104281 | 0.203288 |
| | rel. freq. in valid data | 0.310698 | 0.224879 | 0.139504 | 0.22798 | 0.273152 | 0.25529 | 0.108946 | 0.212381 |

*Emotion Means, Maxima, and Relative Frequencies of Differing Data Points*

**Table A6.** *Cont.*

| | Subject | Neutral | Happy | Sad | Angry | Surprised | Scared | Disgusted | Contempt |
|---|---|---|---|---|---|---|---|---|---|
| | | \multicolumn | | | | | | | |

| | Subject | Emotion Means, Maxima, and Relative Frequencies of Differing Data Points | | | | | | | |
|---|---|---|---|---|---|---|---|---|---|
| | | Neutral | Happy | Sad | Angry | Surprised | Scared | Disgusted | Contempt |
| #55 | Mean | $-1.5 \times 10^{-5}$ | $4.36 \times 10^{-7}$ | $-1.6 \times 10^{-6}$ | $-0.0014$ | $-9.3 \times 10^{-6}$ | $-0.00147$ | $-0.00033$ | $-1.6 \times 10^{-8}$ |
| | maximum | 0.007317 | $5.29 \times 10^{-5}$ | $4.08 \times 10^{-6}$ | 0.114174 | $1.99 \times 10^{-5}$ | 0.143154 | 0.026798 | $4 \times 10^{-8}$ |
| | rel. freq. in all data | 0.059224 | 0.033343 | 0.031944 | 0.033762 | 0.052043 | 0.035208 | 0.029565 | 0.036794 |
| | rel. freq. in valid data | 0.063602 | 0.035807 | 0.034305 | 0.036258 | 0.055889 | 0.03781 | 0.031751 | 0.039513 |
| #65 | Mean | $2.13 \times 10^{-5}$ | $-1.5 \times 10^{-5}$ | $3.97 \times 10^{-5}$ | $-3.5 \times 10^{-6}$ | $-5.7 \times 10^{-5}$ | $1.49 \times 10^{-6}$ | 0.000171 | $-2.3 \times 10^{-8}$ |
| | maximum | 0.012913 | 0.001319 | 0.030275 | 0.003676 | 0.024688 | 0.001189 | 0.02302 | $2 \times 10^{-7}$ |
| | rel. freq. in all data | 0.34951 | 0.024265 | 0.303398 | 0.137343 | 0.180346 | 0.211476 | 0.037304 | 0.219118 |
| | rel. freq. in valid data | 0.500309 | 0.034734 | 0.434302 | 0.196601 | 0.258158 | 0.302719 | 0.053399 | 0.313659 |
| #70 | Mean | $-3.8 \times 10^{-6}$ | $-1 \times 10^{-6}$ | $8.16 \times 10^{-6}$ | $-1.2 \times 10^{-7}$ | $-7.7 \times 10^{-7}$ | $-1.7 \times 10^{-8}$ | $-6.7 \times 10^{-8}$ | $-9.5 \times 10^{-8}$ |
| | maximum | 0.003799 | $3.25 \times 10^{-6}$ | 0.007671 | $7.72 \times 10^{-5}$ | $3.09 \times 10^{-6}$ | $1 \times 10^{-7}$ | $2 \times 10^{-7}$ | $3.53 \times 10^{-7}$ |
| | rel. freq. in all data | 0.08696 | 0.020023 | 0.088946 | 0.073142 | 0.038888 | 0.016465 | 0.030159 | 0.067433 |
| | rel. freq. in valid data | 0.125643 | 0.02893 | 0.128512 | 0.105678 | 0.056186 | 0.02379 | 0.043574 | 0.09743 |
| #94 | Mean | $-2.2 \times 10^{-5}$ | $2.63 \times 10^{-5}$ | $9.85 \times 10^{-7}$ | $2.78 \times 10^{-6}$ | $-5.8 \times 10^{-6}$ | $-5.3 \times 10^{-7}$ | $-4 \times 10^{-6}$ | $-4.9 \times 10^{-7}$ |
| | maximum | 0.020348 | 0.069351 | 0.02824 | 0.005737 | $3.15 \times 10^{-5}$ | $1.81 \times 10^{-6}$ | 0.050832 | 0.000802 |
| | rel. freq. in all data | 0.514396 | 0.25565 | 0.28763 | 0.33138 | 0.331426 | 0.278346 | 0.264607 | 0.322986 |
| | rel. freq. in valid data | 0.801959 | 0.398567 | 0.448425 | 0.516631 | 0.516704 | 0.43395 | 0.41253 | 0.503546 |

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
