# Peer review of "Test–Retest Reliability in Automated Emotional Facial Expression Analysis: Exploring FaceReader 8.0 on Data from Typically Developing Children and Children with Autism"

_applsci, doi:10.3390/app12157759_

Round 1

Reviewer 1 Report

In the present study, the authors used the automated emotional facial expression analysis (AEFEA) in 60 children (31 health and 29 with ASD), in order to generate explore the test-retest reliability of a specific AEFEA software. They conclude that further exploration of test-retest reliability of AEFEA systems is clearly desirable.

The study is well performed, the methods are sound, and the results are robust. I do have some comments to improve the readability of the manuscript.

Introduction: this section is too long,  it should be shrieked and more clarity and conciseness are necessary. I would suggest removing/reducing the paragraph between lines 53-57. Also, in my opinion the paragraph between lines 152-159 should be in the "Methods" section, under "study aims". Also, the paragraph between lines 167-191, should be shrinked/removed and/or placed into the discussion section.

Methods: The term "white" is now preferred to the old  "Caucasian". Also, clarify what does "typically developing" means. 

Figure 3: Please, avoid abbreviations/contract forms in the figure text (x axis). 

Results: this section is too long. I would suggest using supplementary materials.

Discussion: the discussion should be more concise and clear. I would suggest dividing it into five big paragraphs: summary of study results, validity of the results within the context of existing literature, possible clinical relevance, limitations, conclusions and next steps.

Reviewer 2 Report

Summary: This study examined the test-retest reliability of an Automated Emotional Facial Expression Analysis system – FaceReader 8.0 (FR8). The system was tested on videos of neurotypical and neurodivergent children (specifically children diagnosed with ASD). The labels given by FR8 in an initial test were compared to those given in a second test (if there was mismatch between the first and second tests, a third test was run and compared with the second). The authors found high test-retest reliability, although not perfect. This study is novel in that it examines the test-retest reliability of AEFEA systems and is therefore valuable, firstly in highlighting that this has not been done before, and secondly in demonstrating that we cannot assume test-retest reliability with such systems, given that the results showed imperfect reliability.

The overall paper is well written with thorough descriptions of the design and materials. The justification is also clearly outlined. The statistical analyses were reported thoroughly and the findings discussed in depth. I commend the authors for their hard work and can definitely see the value of this work in highlighting the need for assessments of test-retest reliability in such systems, as well as exploration into the causes behind imperfect test-retest reliability. 

General suggestions for improvement:

The Introduction is very thorough and presents all of the relevant information to understand the motivation behind this study. However, I think it would benefit from being slightly restructured to make sure that the last thing being presented in the introduction is the aim of the present study and the research questions. I would, therefore, suggest moving the research questions to after the discussion about the relevance of AEFEA technologies to clinical diagnosis and ASD.

Could the authors please clarify whether the video data required any pre-processing before being fed to the FR8 as input, or is the FR8 able to just take raw video data?

Given that there are a number of human emotional facial expression datasets available, can the authors provide their justification for collecting a novel dataset? Was this due to the fact that they wanted a dataset that included children with ASD, and such a dataset is not readily available?

On a related note, the authors point out that, within the existing literature, it is difficult to compare studies on validity and reliability because of the differences in procedure and stimulus datasets used. However, this study uses a newly constructed dataset. I'm therefore curious whether this dataset will be made available to other researchers and, if so, how it can be accessed? I know the data from this study has been made available but does this include the dataset materials? 

Minor suggestions:

Line 55 – please introduce Nodulus FaceReader 8.0 by it’s full name in the introduction before using the FR8 abbreviation

Line 56 – starting this sentence with “it” is a bit ambiguous. I would suggest clarifying here, for example, “The results of our study don’t question the validity…” or “These findings don’t questions…”

Line 57 – this sentence might flow better if rephrased to something like: “demonstrates that further exploration is needed to better understand this key characteristic of AEFEA systems.”

Line 87 – correction “replaced with data”

Lines 101-115 - I’m a bit unclear on the intended purpose of this paragraph. I think it is presenting a review of evidence about the validity of the FaceReader system. I think my confusion stems from there seeming to be no link between the sentences in lines 97-100, and the start of the paragraph in line 101. It might be worth removing lines 97-100 and clarifying the opening sentence in line 101. For example, “In regards to the FaceReader system specifically, three different versions of FaceReader have been validated on, in total, four different validated datasets of….”

Line 126 – “(see below)” – I’m not sure what I should be seeing below here. If this is just a way to let the reader know that you’ll be discussing how test-retest reliability has been taken for granted in more depth, I would suggest just leaving this comment out.

Lines 139-142 – personally I think this section could be left out. It doesn’t add much to the overall argument

Line 178 – grammatical error = “significantly fewer EFEs”

Line 205 – just for the sake of clarity it might be worth adding to this sentence how many children were in each group. I know this is mentioned later but it would also work to be added here and would mean the reader has the important information up-front. I would also recommend mentioning whether a power analysis was done.  

Line 549 – there’s a bit of ambiguity when you say “are these findings possibly artefacts”. This section would benefit from a more explicit description of what is meant by “these findings”. For example, “is the imperfect test-retest reliability an artefact of how the FR8 was implemented in this study? (2) if not, …..”

Line 567 – what is meant by “data completeness”? How was this measured in order to be compared with other recent studies?
